# Dynamics for El Niño-La Niña asymmetry constrain equatorial-Pacific warming pattern

Michiya Hayashi[1,3 ✉], Fei-Fei Jin[1 ✉] & Malte F. Stuecker [2]

The El Niño-Southern Oscillation (ENSO) results from the instability of and also modulates the strength of the tropical-Pacific cold tongue. While climate models reproduce observed ENSO amplitude relatively well, the majority still simulates its asymmetry between warm (El Niño) and cold (La Niña) phases very poorly. The causes of this major deficiency and consequences thereof are so far not well understood. Analysing both reanalyses and climate models, we here show that simulated ENSO asymmetry is largely proportional to subsurface nonlinear dynamical heating (NDH) along the equatorial Pacific thermocline. Most climate models suffer from too-weak NDH and too-weak linear dynamical ocean-atmosphere coupling. Nevertheless, a sizeable subset (about 1/3) having relatively realistic NDH shows that El Niño-likeness of the equatorial-Pacific warming pattern is linearly related to ENSO amplitude change in response to greenhouse warming. Therefore, better simulating the dynamics of ENSO asymmetry potentially reduces uncertainty in future projections.

[1] Department of Atmospheric Sciences, SOEST, University of Hawaiian ʻOkina at Mānoa, 2525 Correa Rd., Honolulu, HI 96822, USA. [2] Department of Oceanography and International Pacific Research Center, SOEST, University of Hawaiian ʻOkina at Mānoa, 1680 East-West Rd., Honolulu, HI 96822, USA. [3] Present address: Center for Global Environmental Research, National Institute for Environmental Studies, 16-2 Onogawa, Tsukuba 305-8506 Ibaraki, Japan. ✉email: hayashi.michiya@nies.go.jp; jff@hawaii.edu

The El Niño-Southern Oscillation (ENSO) originates from ocean-atmosphere coupled feedbacks in the equatorial Pacific[1–3] and also has a nonlinear rectification effect onto the cold tongue climate state[4–9], affecting global climate and weather. Advances in ENSO theory and modeling have led to improved understanding of ENSO physics. Nevertheless, capturing ENSO's spatio-temporal complexity, as well as the correct balance of coupled feedbacks, remains an outstanding challenge[10,11]. For instance, while most of state-of-the-art ocean-atmosphere coupled climate models simulate its overall spatial pattern and temporal evolution realistically, this seeming realism occurs often for the wrong reasons. Large error cancellations are evident in the majority of climate models between positive coupled dynamic- and negative thermodynamic feedback processes that determine ENSO dynamics[12–15]. Furthermore, most models still fail to reproduce ENSO nonlinearity such as the observed asymmetry of sea surface temperature (SST) anomalies in the eastern equatorial Pacific[16–18]. These deficiencies are likely able to explain discrepancies among previous studies investigating ENSO properties in response to a changing climate[19–28]. Thus, the important question of which key factors control the diversity in simulated ENSO asymmetry among climate models remains a subject of debate[13,14,16–18,29].

Asymmetry in SST anomalies between warm El Niño and cold La Niña phases is often measured by the normalized third statistical moment, i.e., skewness[30]. In the eastern Pacific, where the ENSO signal is the strongest, the observed SST (and subsurface ocean temperature) skewness is highly positive (Fig. 1a, b) due to the presence of extreme El Niño events and typical absence of as extreme La Niña events. This ENSO asymmetry is known to result in a residual warming signal, rectifying to warmer mean-state ocean temperatures in the eastern equatorial Pacific[4–9]. However, ENSO asymmetry is poorly reproduced in most climate models[13,16,31]. The poor simulation of ENSO skewness also undercuts the models' ability to simulate a realistic occurrence percentage of extreme El Niño events[32], potentially reducing the nonlinear rectification effect onto the climate mean state[7,16,25,33]. Possible causes for this deficiency were previously suggested to be related to the Pacific mean-state SST bias that tends to anchor the atmospheric Walker Circulation too far westward, which may affect atmospheric feedbacks[7,13–15,33–35], and state-dependent noise ENSO excitation[32,36]. However, the question of what dominant nonlinear dynamical process is causing it remains elusive.

The oceanic nonlinear dynamical heating (NDH) is a deterministic advective process that enhances ENSO asymmetry[5,6,37]. Although earlier studies focused on the NDH in the surface mixed layer[8,9,16], ENSO's nonlinear behavior is also prominent in the subsurface ocean[5,7,9,33]. Below the mixed layer, an intense mean eastward current exists along the equator in the eastern Pacific, called the Equatorial Undercurrent (EUC). It is caused by a mean zonal pressure gradient force that is maintained by the easterly trade winds and westward surface current[38,39]. It is observed that the EUC is weakened or halted by anomalous central-Pacific westerly winds during strong El Niño events[5], generating the intense NDH along the equatorial Pacific thermocline (Fig. 1c, d). A recent observational study revealed that this subsurface NDH substantially reduces the cooling subsurface temperature tendency in the transition phase toward La Niña, enhancing ENSO asymmetry[37]. However, no study has investigated whether the current generation of climate models can simulate the subsurface NDH of ENSO realistically.

This study aims to evaluate the subsurface NDH and ENSO asymmetry as well as atmospheric nonlinearities associated with ENSO feedbacks in state-of-the-art climate model simulations and to detect consequences thereof for future climate projections through constraining a climate model ensemble. Using multiple reanalysis datasets and climate model outputs for the Coupled Model Intercomparison Project Phase 5 (CMIP5; ref. [40]) and Phase 6 (CMIP6; ref. [41]), we show that ENSO asymmetry in CMIP models is largely proportional to the intensity of simulated subsurface NDH. Our results suggest that simulating the dynamics associated with ENSO asymmetry correctly is a necessary condition to constrain the tropical warming pattern due to ENSO amplitude change in a changing climate.

## Results

**Simulated ENSO asymmetry and nonlinear dynamical heating.** The positive skewness of the eastern-Pacific temperature anomalies that characterizes ENSO asymmetry is poorly reproduced among 25 CMIP5 and 26 CMIP6 historical simulations (Supplementary Table 1), despite the fact that simulated ENSO SST amplitude is in a reasonable range compared to observations (Fig. 1e, f). Figure 2a shows that the standard deviation of detrended Niño-3 SST anomalies ($\sigma_{ENSO}$; 150°–90°W, 5°S–5°N) is very close to the observations on average, however, the multi-model mean of the skewness ($\gamma_{ENSO}$) cannot be statistically distinguished from zero. Thus, CMIP climate models fail badly in simulating ENSO skewness that is about 1 in observations, indicating no improvement in CMIP6 compared to earlier CMIP phases[16–18,31].

Here we show that ENSO SST skewness is highly constrained by subsurface NDH variability in climate models, using multiple reanalysis datasets and the subset of 25 CMIP5 and 18 CMIP6 historical simulations available for evaluating NDH (not all models provide all the necessary ocean data fields that are required to calculate NDH—see "Methods" section and Supplementary Table 1). The subsurface NDH is derived from the nonlinear temperature advective terms in the equatorial eastern-Pacific box (100°W–180°, 1°S–1°N, 50–150 m)[37], where an intense NDH gives rise to rectified warming along the mean thermocline and EUC in reanalysis (Fig. 1c, d). In the CMIP climate models, both the mean and variability of the subsurface NDH are too weak (Fig. 1g, h), indicating serious deficiencies in simulating subsurface ocean nonlinear dynamics.

The level of ENSO asymmetry ($\gamma_{ENSO}$) linearly increases with respect to the relative strength of subsurface NDH variability to ENSO amplitude ($\sigma_{NDHsub}/\sigma_{ENSO}$; referred to as NDH efficiency) among the CMIP models with a correlation coefficient of 0.78 ($p < 0.00001$; Fig. 2b). This explains ~60% of the model-to-model $\gamma_{ENSO}$ variance. However, only a few models have an NDH efficiency that is comparable to the observations at 0.30 month$^{-1}$ on average (Fig. 2b). In contrast, the inter-model spread in surface NDH above 50 m depth does not explain the spread in $\gamma_{ENSO}$ (Supplementary Fig. 1). This is supported by an analytical nonlinear ENSO model[42], which indicates that surface NDH does not guarantee to enhance the positive ENSO SST skewness. Therefore, we hypothesize that the inability to simulate subsurface NDH is a key dynamical factor responsible for the lack of ENSO asymmetry in climate models.

In addition to the abovementioned deficiencies in the subsurface NDH efficiency, weak ENSO asymmetry in climate models can be attributable to atmospheric nonlinearities in the dynamic and thermodynamic feedback processes of ENSO[13,14,25,33,35,43,44]. A previous study[14] indicates that the El Niño-La Niña asymmetry in shortwave (SW) surface heat flux feedback, characterized by an east-west contrast pattern near the dateline in association with the location of the main convection region, is underestimated in most climate models. Thus, we examine the zonal contrast of equatorial Pacific SW anomalies (140°–170°E minus 140°–170°W, 5°S–5°N) in terms of its regression coefficient difference for positive and

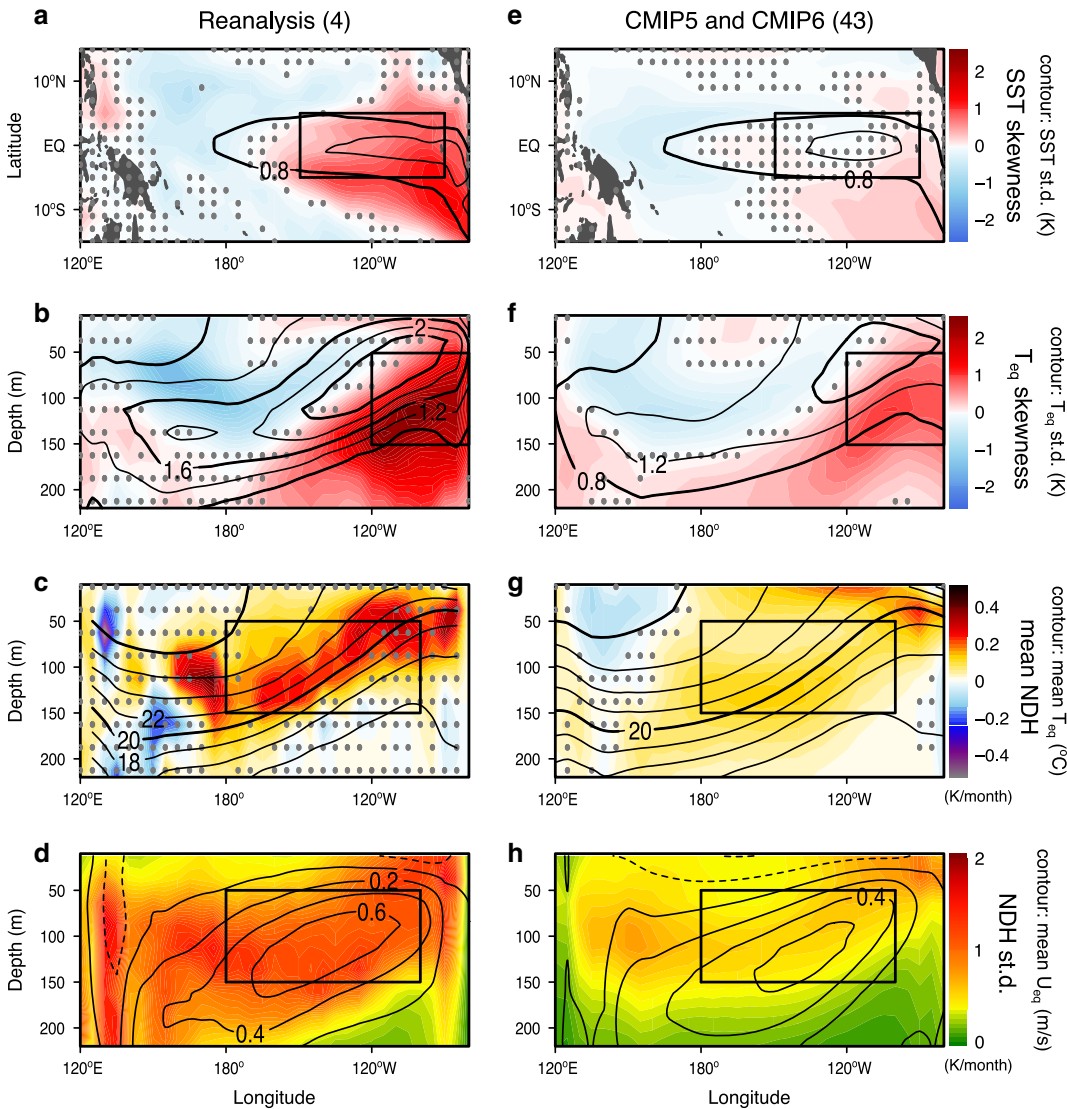

**Fig. 1 Nonlinearity in the equatorial Pacific Ocean. a** Horizontal map of the skewness $\gamma$ (shading) and standard deviation $\sigma$ (contours, K) of the sea surface temperature (SST) anomaly. **b** Equatorial cross section of $\gamma$ (shading) and $\sigma$ (contours, K) of the potential temperature anomaly. **c** The long-term mean of nonlinear dynamical heating (NDH) (shading, K month$^{-1}$) and potential temperature (contours, °C). **d** $\sigma$ of the NDH (shading, K month$^{-1}$) and the long-term mean of the zonal ocean current (contours, m s$^{-1}$). Four reanalysis datasets are used in (**a–d**). Dots in **a–c** indicate the shading values that are not statistically significant at the 95% confidence level. Boxes represent the Niño-3 region in (**a**) while the averaging regions for the mean and $\sigma$ of NDH in (**c**, **d**) and the skewness of subsurface temperature in (**b**). **e–h** Same as in (**a–d**), except for the historical simulations of 43 CMIP models.

negative Niño-3 SST anomalies ($\Delta_x$SW feedback asymmetry) in Fig. 2c. Indeed, the simulated ENSO SST skewness tends to increase with respect to $\Delta_x$SW feedback asymmetry with a moderate correlation coefficient of 0.37 ($p = 0.008$). However, the simulated ENSO skewness is generally too low even in the climate models that have higher feedback asymmetry. We also confirm that the dynamic feedback asymmetry derived from the zonal wind stress anomalies in the central-Pacific region (CP; 150° E–120°W, 5°S–5°N) is nearly zero in reanalysis datasets despite that most climate models simulate positive asymmetry in the dynamic feedback (i.e., stronger wind feedback for El Niño than La Niña; Fig. 2d). Thus, it does not appear that enhancing either the SW- or wind feedback nonlinearity would help to considerably improve simulated ENSO SST skewness, calling a need for other sources of the nonlinearity. Therefore, we hereafter focus on the subsurface NDH as a dynamic nonlinear source for simulating ENSO asymmetry.

**Model biases in the dynamics of ENSO asymmetry.** To detect the dynamics responsible for models' common biases that prevent simulating realistic ENSO asymmetry, we classify CMIP into subgroups (Table 1) using the inter-model fidelity in the subsurface NDH efficiency (Fig. 2b). The group H is composed of 10 CMIP5 and 4 CMIP6 models that have higher levels of NDH efficiency than the multi-model mean (0.16 month$^{-1}$) and are thus closer to the observations (0.30 month$^{-1}$). Seven group H models with an NDH efficiency even greater than 0.20 month$^{-1}$ are further classified as a subset, group HH. The other 29 models are categorized as group L (low NDH efficiency). Figure 3 shows the simulated nonlinearity in each model group in terms of the SST skewness and mean equatorial NDH (see also Supplementary Fig. 2). In group H, the skewness is broadly positive in the eastern- and negative in the western Pacific similar to the reanalysis (Fig. 1a, b), except that simulated positive skewness is still too weak. The group HH models perform even better (Fig. 3c).

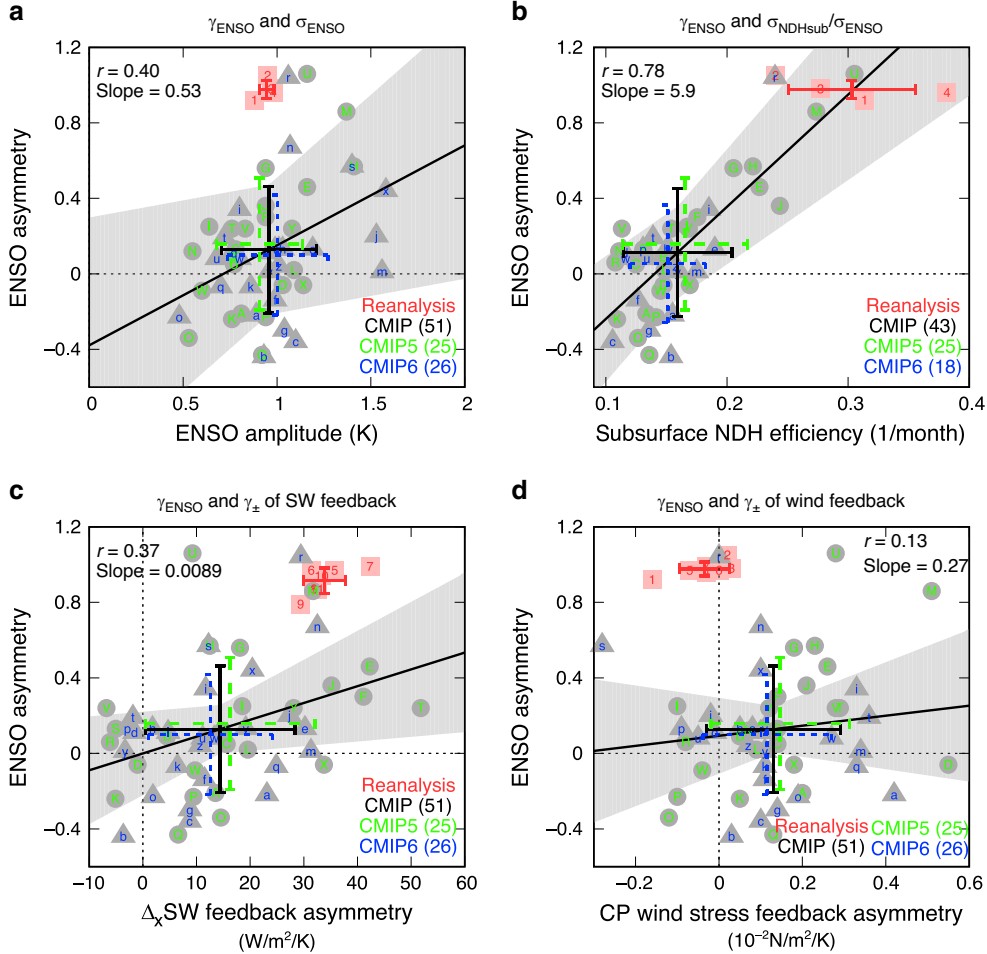

**Fig. 2 ENSO asymmetry and nonlinear processes.** Skewness of sea surface temperature (SST) anomalies in the Niño-3 region as functions of ENSO amplitude (**a**), the efficiency of subsurface nonlinear dynamical heating (**b**), feedback asymmetry of the zonal contrast ($\Delta_x$; 140°–170°E minus 140°–170° W, 5°S–5°N) of shortwave anomalies (**c**), and feedback asymmetry of central-Pacific (CP; 150°E–120°W, 5°S–5°N) zonal wind stress anomalies (**d**). In (**c**, **d**) feedback asymmetry ($\gamma_\pm$) is the difference between regression coefficients for positive and negative Niño-3 SST anomalies. Error bars denote the one standard deviation range for all models (black), CMIP5 (green with circles), CMIP6 (blue with triangles), and reanalysis (red with squares). In each bracket, the number of available models is indicated. The linear fitting lines for all the models are shown with shading for the 95% confidence ranges of slopes and intercepts.

**Table 1 Brief summary of CMIP model availability in each group.**

| Group | #CMIP | #CMIP5 | #CMIP6 | Description |
|---|---|---|---|---|
| | 43 (51)[a] | 25 (25) | 18 (26) | Models available for evaluating ENSO's SST and feedbacks |
| | 38 (43) | 25 (25) | 13 (18) | Models available for evaluating the subsurface NDH |
| L | 25 (29) | 15 (15) | 10 (14) | Models having low NDH efficiency |
| H | 13 (14) | 10 (10) | 3 (4) | Models having high NDH efficiency |
| HH | 7 (7) | 6 (6) | 1 (1) | Group H models having even higher NDH efficiency |
| H-sEN | 7 | 5 | 2 | Subgroup of H models projecting a strengthening ENSO |
| H-wEN | 6 | 5 | 1 | Subgroup of H models projecting a weakening ENSO |

*SST* sea surface temperature, *NDH* nonlinear dynamical heating.
[a]The brackets show the number of models available for the historical simulations.

Furthermore, as seen in the reanalysis (Fig. 1c, d), the positively skewed intense NDH variability near the thermocline results in positive long-term mean residuals (Fig. 3e, f)—a rectification onto the climate mean state[9,37]. In contrast, the group L models fail to reproduce these key nonlinear properties (Fig. 3a, d).

The group L models, consisting of about 70% of the available CMIP ensemble but severely suffering from poor ENSO nonlinearity, show discernible differences from the reanalysis

datasets (Fig. 4). In the composited mean states of group L, the eastern-Pacific cold tongue extends far to the west[13–15,18,34,45,46] (Supplementary Fig. 3), accompanied by a too intense westward ocean surface current. Nevertheless, the simulated amplitudes of mean EUC and easterly trade winds are comparable with the reanalysis[39] (Supplementary Fig. 4). In contrast, the anomalous CP zonal wind stress response to ENSO SST anomalies is too weak (Fig. 5a) even though the SST anomaly amplitude and

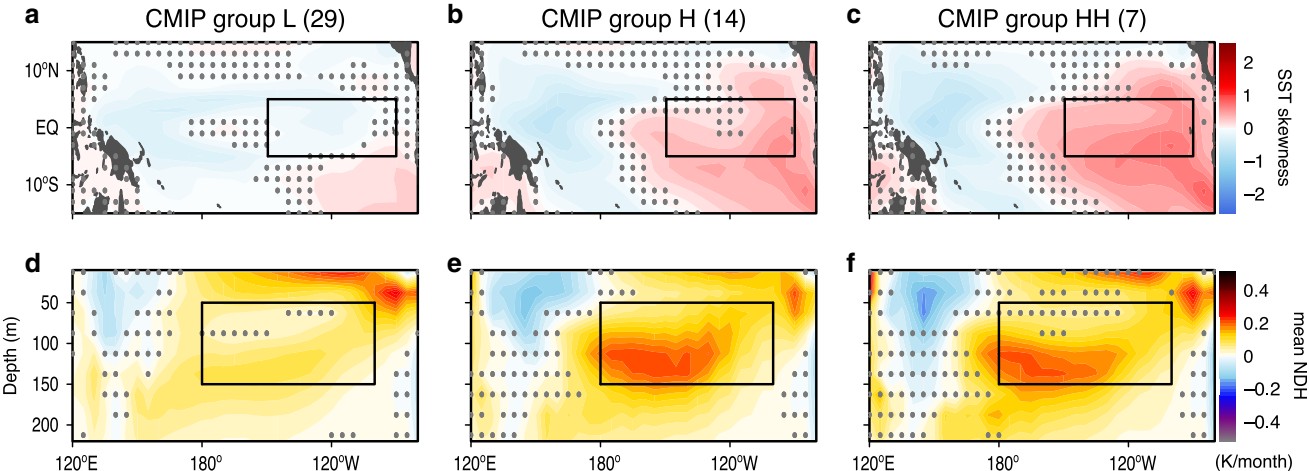

**Fig. 3 Simulated SST skewness and equatorial mean NDH.** Composite structures of (**a–c**) the skewness of the sea surface temperature (SST) anomaly and (**d–f**) the long-term mean of nonlinear dynamical heating (NDH; K month$^{-1}$) in the historical simulations for each model group. Dots indicate the shading values that are not statistically significant at the 95% confidence level. Boxes represent the Niño-3 region in **a–c** while the averaging regions of NDH in (**d–f**).

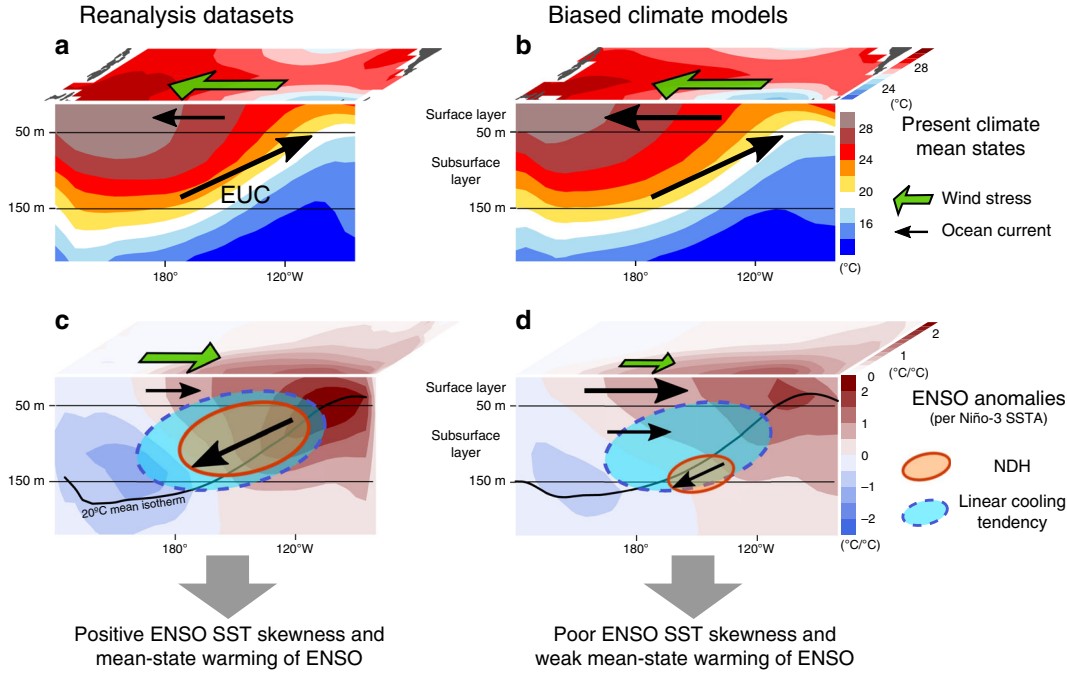

**Fig. 4 Schematic of oceanic nonlinear dynamics for ENSO asymmetry.** Shown are the climate mean states (**a**, **b**) and ENSO anomalies (**c**, **d**) for the sea surface temperature from the equator to 15°N and equatorial potential temperature derived from the multiple reanalysis datasets (**a**, **c**) and a subset of CMIP climate models (group L) having too-weak nonlinear dynamical heating (NDH) efficiency (**b**, **d**). Green and black arrows represent the zonal wind stress and ocean currents along the equator. Solid orange and dashed cyan ovals indicate the NDH and linear dynamical (advective) cooling tendency, respectively, in the transition phase from El Niño to La Niña. See Supplementary Figs. 2 and 4 for the details of each plot.

precipitation response are close to observations (Supplementary Figs. 2 and 4). In addition to this SST-wind coupling bias, most group L models fail to reproduce the anomalous westward EUC response to westerly wind anomalies over the central Pacific[39] (Fig. 5b). Indeed, the zonal current covarying with ENSO is less intense in the subsurface but too strong in the western-Pacific surface layer (Supplementary Fig. 4). These two biased linear dynamical coupling processes—from SST to winds and from winds to the EUC—can prevent the generation of wind stress induced subsurface NDH that would normally enhance positive skewness of subsurface temperature and SST anomalies in the

eastern equatorial Pacific[37] (Fig. 4c, d; see also Supplementary Fig. 5).

In the group H models that better simulate the subsurface NDH efficiency and ENSO SST skewness, these two linear coupling processes (both from SST to winds and from winds to EUC) are improved (Fig. 5) and also the excessive mean cold-tongue bias tends to be reduced (Supplementary Fig. 3). The linear wind-EUC coupling is further improved in group HH, which is a subset of the group H models that have higher NDH efficiency. A common bias still exists in groups H and HH with negative mean NDH in the western-Pacific surface layer (Fig. 3d–f), potentially due to too

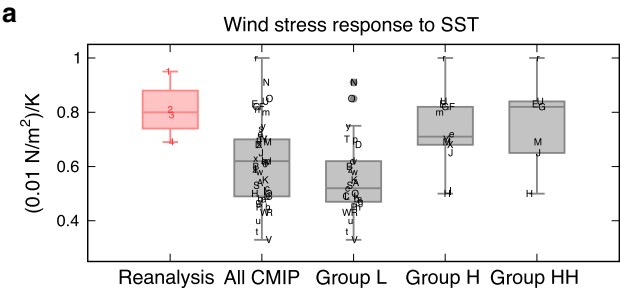

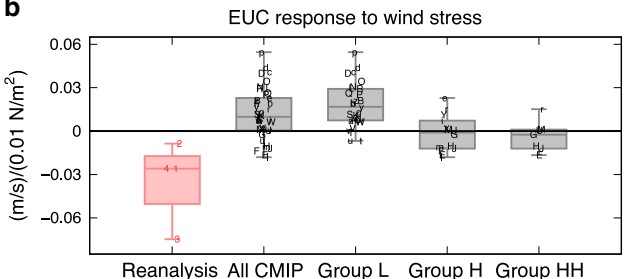

**Fig. 5 Simulated linear atmosphere-ocean coupling.** The central-Pacific (CP; 150°E–120°W, 5°S–5°N) zonal wind stress anomalies regressed onto the Niño-3 sea surface temperature anomalies (**a**) and the subsurface zonal current anomalies beneath the Niño-3 region (150°–90°W, 1°S–1°N, 50–150 m depth) regressed onto the CP zonal wind stress anomalies (**b**). Shown are the medians (horizontal lines) and first and third quartiles (boxes) for the reanalysis datasets, all the CMIP5 and CMIP6 climate models, and the models in groups L, H, and HH. Letters indicate each reanalysis or climate model. Samples outside of 1.5 times the interquartile range from each box are plotted with the closed circles as outliers (models "O" and "N" of group L in **a**).

strong anomalous eastward ocean currents associated with El Niño and a too strong mean-state westward surface current (Fig. 4 and Supplementary Fig. 4). Nevertheless, the group H models perform noticeably better in simulating oceanic nonlinear dynamics as well as linear ENSO feedback processes.

The majority of the climate models fail to simulate the intensity of the linear coupled dynamic feedbacks from SST to winds and from winds to the EUC (Fig. 5). As previous studies suggested[12–15], the errors in the SST-winds coupling tend to be compensated by errors in thermodynamic radiative feedbacks, resulting in a seemingly realistic growth rate and thus ENSO amplitude (Supplementary Fig. 6). However, this error compensation falls apart for ENSO asymmetry because the asymmetry is largely affected by errors in the dynamical coupling alone through the subsurface NDH. Thus, simulating both linear dynamical coupling and thermodynamic feedback correctly (i.e., reducing error compensations that have occurred in climate models) shall improve ENSO asymmetry. It will be important for modeling ENSO nonlinearity to elucidate why the wind stress response to ENSO SST anomalies is too weak and what prevents a realistic EUC response to a given wind forcing.

**Constrained tropical response to ENSO amplitude change.** The ENSO nonlinearity is a potential source that modulates the current climate mean state[4–6,9,16,33] and future projections[7,23,25,26]. However, we have shown in the previous section that the majority of the CMIP climate models fail to reproduce the dynamics of ENSO asymmetry. Thus, we next address whether the future tropical climate response is related to ENSO amplitude change if we constrain the CMIP climate model ensemble to the L, H, and

HH subsets. Hence, we use a dynamics-oriented criterium for ENSO asymmetry rather than one based only on SST statistics as is done in previous studies[25,26] to hopefully provide a clearer illustration of the consequences of ENSO changes through non-linear dynamics.

As the group H models are able to simulate the essential properties of subsurface NDH, we expect that in these models an increase of future ENSO amplitude should enhance the mean-state rectification effect that warms the equatorial eastern-Pacific subsurface and deepens the thermocline. In contrast, a future reduction of ENSO amplitude in these models should lead to less rectified warming in the equatorial eastern-Pacific subsurface. Importantly, we do not expect this effect for models that do not realistically simulate NDH (group L). Comparing the historical period to the future scenario simulations for 2051–2100 (RCP8.5 for CMIP5 and SSP5-8.5 for CMIP6; see "Methods" section), we find that the future change in the long-term mean of the subsurface NDH linearly follows ENSO amplitude change in group H (Fig. 6a). However, the projected ENSO amplitude change shows still a large spread even within group H: 7 of these models project a strengthening of ENSO (group H-sEN), while the other 6 models a weakening (group H-wEN). Quantifying ENSO changes in a future warm climate remains a challenge[26–28] as it requires climate models to simulate ENSO processes realistically in the current climate[20]. As for the group H-wEN models at least (Supplementary Fig. 7), we confirm that the weakening of ENSO is attributable to the declining oceanic wave response to equatorial-Pacific zonal winds due to an intensified thermal stratification[24]. Nevertheless, the spread of ENSO amplitude change in group H does give us an opportunity to address the question of how ENSO asymmetry and amplitude together can affect subsurface NDH and thus ENSO's nonlinear rectification onto the climate mean state.

What kind of surface warming pattern will emerge in the equatorial Pacific in response to greenhouse gas forcing is still a subject of active debate[20,23–25,27,39,47–52]. Here we show that the projected warming pattern is strongly related to ENSO amplitude change once climate models are conditioned by their fidelity in dynamics responsible for ENSO asymmetry (as reflected in the simulated NDH efficiency). A measure of El Niño-likeness of tropical Pacific warming, defined as the spatial correlation coefficient of the SST trend pattern in the scenario simulations with the ENSO-regressed SST anomaly pattern in the historical simulations over 90°E–60°W and 20°S–20°N (see "Methods" section), is positive (i.e., more warming in the east than the west) in most CMIP models[20,25,26,48] but highly uncertain ranging from −0.11 to 0.83 (Fig. 6b). In group H, we find a statistically significant increase of the El Niño-likeness as ENSO amplitude increases ($r = 0.57$, $p = 0.042$). This relationship is strengthened to a correlation coefficient of 0.89 ($p = 0.0073$) within the group HH models that have greater NDH efficiency. The El Niño-likeness in group L is positive, ranging from 0.33 to 0.73, and there exists no correlation with ENSO amplitude change ($r = 0.00$) as expected because a simulated ENSO without realistic NDH is not able to yield a noticeable rectification effect.

How the projected ENSO change affects future trends is detectable in the difference between the strengthening and weakening ENSO (sEN and wEN) simulations once the CMIP ensemble is conditioned by the fidelity to simulated ENSO nonlinear dynamics (Fig. 7). In a CMIP5 greenhouse warming scenario (RCP8.5), the tropical-Pacific surface warming trends of groups H-sEN and H-wEN are significantly different while those of groups L-sEN and L-wEN cannot be distinguished statistically (Fig. 7c, i). For instance, the surface warming in the eastern equatorial Pacific is enhanced in H-sEN whereas it is reduced in H-wEN (Fig. 7a, b), yielding more eastern-Pacific warming in

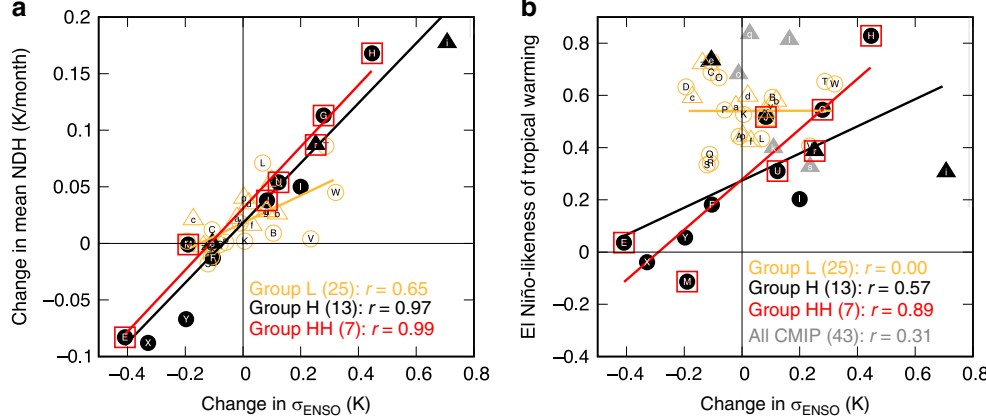

**Fig. 6 ENSO amplitude change and tropical warming pattern. a** Future change in the long-term mean of subsurface nonlinear dynamical heating (NDH) as a function of ENSO amplitude change. **b** El Niño-likeness of the tropical Pacific warming as a function of ENSO amplitude change. Triangles and circles indicate the CMIP6 and CMIP5 models, respectively, in group H (black, closed) and group L (orange, open). Red open squares represent the group HH models. Gray closed triangles represent 5 CMIP6 models unavailable for evaluating NDH variability (Table 1; see "Methods" section for details).

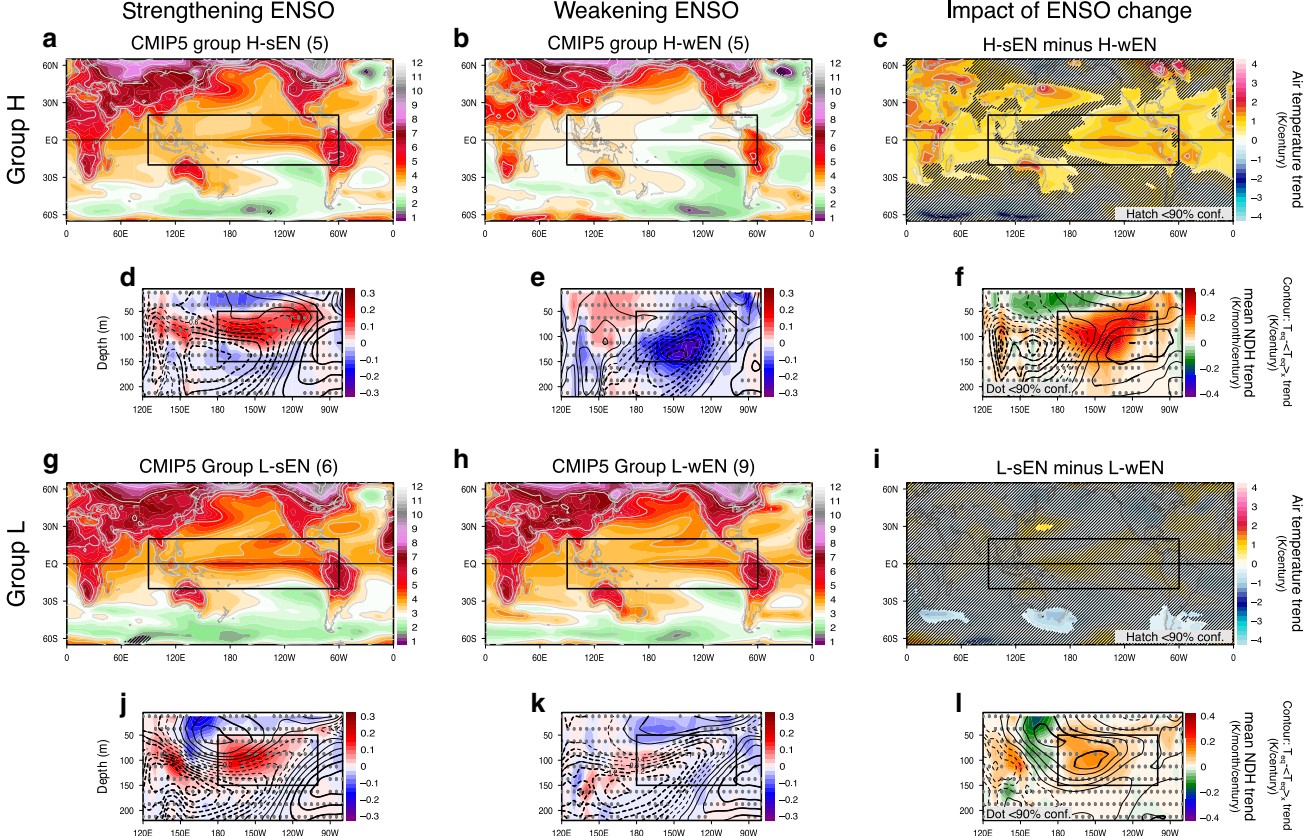

**Fig. 7 Surface warming due to ENSO amplitude change. a–f** Linear trends of the near-surface air temperature (**a–c**) and equatorial nonlinear dynamical heating (NDH) (**d–f**) in groups H-sEN and H-wEN and their difference (H-sEN minus H-wEN) in the CMIP5 Representative Concentration Pathway 8.5 scenario simulations. In **d–f** the contours show the departure of temperature trend for 1°S–1°N from its Pacific zonal mean between 120°E and 80°W. **g–l** As in (**a–f**) but for groups L-sEN and L-wEN. Hatched and dotted areas over shading indicate that the values are not statistically significant at the 90% confidence level. Thick contours indicate statistically significant values at the 90% confidence level. Solid boxes represent the regions for the El Niño-likeness calculation of the warming trend and the NDH average.

their difference as the response to increasing ENSO amplitude (Fig. 7c). A difference between these two groups is also discernible in the Atlantic and Indian Oceans, as well as many land regions in East Asia, Africa, Australia, South America, and Europe. Correspondingly, precipitation increases more in the eastern equatorial Pacific for H-sEN but in the western Pacific for H-

wEN (Supplementary Fig. 8a–c). In contrast, the group L models project El Niño-like tropical warming only regardless of ENSO amplitude change (Figs. 6b and 7g–i). The mean NDH change following the ENSO amplitude change in group H (Fig. 6a) leads to an east-west contrast of subsurface warming difference between H-sEN and H-wEN (Fig. 7d–f) while the mean NDH

change is small in group L so that ENSO-related trends are barely discernible (Fig. 7j–l).

The different responses to ENSO amplitude change between groups H and L reveal the importance of constraining the CMIP ensemble by their fidelity of realistically simulating both linear and nonlinear ENSO dynamics for detecting the ENSO mean-state rectification effect. Only if nonlinear ENSO dynamics are captured realistically in climate models, enhancing of the subsurface rectification effect (that warms the eastern-Pacific subsurface temperature, deepens the mean-state thermocline, and thereby increases eastern-Pacific SST) can lead to a more pronounced El Niño-like global-warming pattern, whereas suppressing the rectification effect leads to less eastern Pacific warming. Without simulating ENSO asymmetry and relevant dynamics realistically, however, climate models project an El Niño-like warming pattern only[50]. This can be attributed to ENSO-independent processes such as reduced equatorial-Pacific mean upwelling due to a weakening of the Walker and Hadley circulations[48,52,53], of which the latter is more prominent in group L (Supplementary Fig. 9). This mean state change might in turn alter ENSO properties[19–22,26,54–56], but ENSO amplitude change shows no correlation to the warming pattern in group L (Fig. 6b).

## Discussion

Here we showed that simulated subsurface NDH is a key controlling factor for ENSO asymmetry among CMIP climate models and the projected future equatorial-Pacific warming pattern. Too weak NDH variability in most climate models stems from biases in linear dynamical ocean-atmosphere coupling processes from SST to winds[13–15,33–35,50] and from winds to ocean currents[39] (Fig. 4). Although the error compensation between linear positive coupled dynamic- and negative thermodynamic feedback processes can result in seemingly reasonable ENSO amplitude[13–15], its crucial weakness is exposed by low-biased ENSO asymmetry. This error compensation hides the relationship between ENSO amplitude change and the equatorial-Pacific warming pattern in response to greenhouse gas forcing unless climate models are conditioned by their fidelity in simulating the important dynamics responsible for ENSO asymmetry. Importantly, the amplitude and pattern of future equatorial-Pacific warming may further affect mean state changes in the broader tropics and beyond via adjustments of the large-scale circulation, thereby potentially constraining global climate sensitivity as well[57]. Therefore, we suggest that realistic climate model simulations of ENSO dynamics are crucial not only for a better understanding of the ENSO phenomenon itself but also for reducing the large uncertainty in projected changes of both tropical and global climate.

## Methods

**CMIP5 and CMIP6 simulations.** We analyzed the historical simulations from 25 CMIP5 models for 1850–2005 (ref. [40]) and 26 CMIP6 models for 1850–2014 (ref. [41]) for the monthly fields of SST (tos in the data archives), zonal wind stress (tauu), near-surface temperature (tas), surface heat fluxes (rsus, rsds, rlus, rlds, hfss, hfls), and precipitation (pr). The potential temperature (thetao) and three-dimensional ocean currents (uo, vo, wo or wmo) were available for all the 25 CMIP5 models and for 18 out of the 26 CMIP6 models for the historical simulations. For the CMIP5 models and 3 of the CMIP6 models (ACCESS-CM2, ACCESS-ESM1-5, BCC-CSM2-MR), the vertical velocity was not provided and thus yielded as the division of the vertical mass transport (wmo) by the horizontal area of a grid cell (areacello) and by the reference density (1035 kg m⁻³) (ref. [58]). The horizontal current vectors of the MPI models were corrected following Shigemitsu et al.[59] since these models have curved equatorial axes. We interpolated the surface variables to a regular 1° × 1° longitude-latitude grid and the ocean variables to a 1° × 0.5° grid. We also used two future scenario simulations with rising radiative forcing pathway leading to 8.5 W m⁻² in 2100: a CMIP5 Representative Concentration Pathway (RCP8.5) for 2006–2100 and a CMIP6 Shared Socio-economic Pathway 5 (SSP5-8.5) for 2015–2100. Since the greenhouse gas emission

pathways are different between these two scenarios[60] and the number of group H models available for the CMIP6 SSP5-8.5 scenario is limited to only 3 (Table 1), the CMIP5 RCP8.5 scenario simulations were used for compositing linear trends (Fig. 7 and Supplementary Figs. 8 and 9). The anomalies of the model outputs were detrended by subtracting the linear trends for each data period based on linear least-square fitting. See Supplementary Table 1 for the data availability in each model.

**Oceanic reanalysis datasets.** We used four reanalysis products for the potential temperature, three-dimensional ocean currents, wind stress, and net surface heat flux: The Ocean Reanalysis System 3 (ORAS3) for 1959–2011 (ref. [61]), the Ocean Reanalysis System 5 (ORAS5) for 1979–2017 (ref. [62]), version 3.3.1 of the Simple Ocean Data Assimilation ocean/sea ice reanalysis (SODA331) for 1980–2015 (ref. [63]), and the National Centers for Environmental Prediction Global Ocean Data Assimilation System (GODAS) for 1981–2017 (ref. [64]). The SST was derived from the nearest-surface potential temperature of each reanalysis dataset.

**Atmospheric reanalysis and observational datasets.** We used three atmospheric reanalysis products together with four oceanic reanalysis datasets for the wind stress and each component of the surface heat flux: ERA-interim for 1979–2018 (ref. [65]), ERA5 for 1979–2018 (ref. [66]), and TropFlux for 1979–2017 (ref. [67]). The radiative surface heat flux components were also derived from OAFlux for 1984–2009 (ref. [68]), CERES EBAF Ed4.0 for 2001–2017 (ref. [69]), GEWEX SRB version 3 for 1984–2007 provided by the NASA Langley Research Center Atmospheric Sciences Data Center NASA/GEWEX SRB Project, and ISCCP-FH for 1984–2009 provided by ISCCP H-series cloud data from NOAA/NCEI. The precipitation was obtained from GPCP version 2.3 for 1979–2018 (ref. [70]). To calculate the regression for these datasets, SST from NOAA ERSST version 5 (ref. [71]) was used for each data period.

**Definitions of the indices.** The ENSO properties are characterized by the detrended monthly Niño-3 SST anomaly (90°–150°W, 5°S–5°N) to assess the variability in the eastern Pacific region, where the SST variability is the most intense in observations and climate models (Fig. 1a, e). The asymmetry in anomalous fields was measured by the normalized statistical third moment, skewness $\gamma = \frac{1}{N}\sum_{i=1}^{N} T_i'^3/\sigma^3$, where $T_i'$ denotes an anomalous field (e.g., SST) at a timestep $i$, $N$ the number of monthly timesteps, and $\sigma = \sqrt{\frac{1}{N}\sum_{i=1}^{N} x_i'^2}$ the standard deviation. We defined ENSO's amplitude and asymmetry as the standard deviation and skewness of the detrended Niño-3 SST anomalies (unit: K). The zonal wind stress anomaly is averaged over a central-Pacific domain (150°E–120°W, 5°S–5°N; unit: N m⁻²). The shortwave- and longwave radiative surface heat flux anomalies are averaged over the Niño-3 and Niño-4 regions (160°E–90°W, 5°S–5°N; unit: W m⁻²). The zonal contrast of shortwave heat flux ($\Delta_x$SW) used in Fig. 2c is the difference between the shortwave anomalies in the western (140°–170°E) and eastern (140°–170°W) boxes between 5°S and 5°N. ENSO feedback is defined as the regression coefficient of a specific anomalous field onto the Niño-3 SST anomaly and the feedback asymmetry is measured by the difference between regression coefficients for positive and negative Niño-3 SST anomalies. The NDH consists of the nonlinear advective terms in the following equation:

$$\frac{\partial T'}{\partial t} = \underbrace{-u'\frac{\partial \bar{T}}{\partial x} - v'\frac{\partial \bar{T}}{\partial y} - w'\frac{\partial \bar{T}}{\partial z} - \bar{u}\frac{\partial T'}{\partial x} - \bar{v}\frac{\partial T'}{\partial y} - \bar{w}\frac{\partial T'}{\partial z}}_{\text{Linear advective terms}}$$
$$\underbrace{-u'\frac{\partial T'}{\partial x} - v'\frac{\partial T'}{\partial y} - w'\frac{\partial T'}{\partial z}}_{\text{NDH}} + res$$

where $T$ indicates the potential temperature; $u$, $v$, and $w$ are the zonal, meridional, and vertical ocean currents; the overbar and prime denote the climatological monthly mean and the detrended anomaly. The residual term (res) includes thermodynamic and subgrid-scale processes, but the total advective terms are dominant for determining the subsurface temperature tendency at least in ocean reanalysis products[37]. Following Hayashi and Jin[37], the subsurface NDH is defined as the NDH terms averaged for 100 W°–180°, 1°S–1°N and 50–150 m depth (Fig. 1c, d). Its monthly standard deviation characterizes the amplitude of the NDH variability (unit: K month⁻¹). The relative amplitude of the NDH variability to ENSO amplitude is called the NDH efficiency (unit: month⁻¹), ranging from 0.24 to 0.38 with an average of 0.30 in four ocean reanalysis products (Fig. 2b).

**ENSO amplitude and El Niño-likeness under global warming.** The future changes of the ENSO amplitude and the long-term mean of NDH are increments from the historical data periods to RCP8.5 or SSP5-8.5 scenario simulations for 2051–2100. Here, the period of the scenario simulations was selected given that the ENSO response to global warming is transient in earlier periods[22–24]. Then, we defined El Niño-likeness as the spatial correlation between the linear SST trend in the scenario simulations and the detrended SST anomalies regressed to the detrended Niño-3 SST anomalies in the historical simulations over the tropical Pacific domain (90°E–60°W, 20°S–20°N; solid box in Fig. 7a–c, g–i). The positive El

Niño-likeness index indicates the so-called El Niño-like warming pattern (enhanced surface warming in the east) of the equatorial Pacific Ocean while La Niña-like warming is characterized by enhanced surface warming in the west.

**Statistical significance test**. The statistical confidence levels were tested by a two-tailed Student's *t*-test for the composited fields and correlation values, except for the differences between two CMIP groups for which the Welch's *t*-test was used (Fig. 7c, f, i, l and Supplementary Figs. 8c, f and 9c). The confidence levels are described in the figure captions.

## Data availability

The source data underlying Figs. 1–7 and Supplementary Figs. 1–9 are provided as a Source data file. The CMIP5 and CMIP6 datasets are publicly available at https://cmip.llnl.gov/ and the CEDA data archive http://data.ceda.ac.uk/. The ORAS5 and ORAS3 datasets are publicly available at http://icdc.cen.uni-hamburg.de/thredds/catalog/ftpthredds/EASYInit/oras5/catalog.html and http://icdc.cen.uni-hamburg.de/thredds/catalog/ftpthredds/EASYInit/ORA-S3/catalog.html, the SODA331 dataset at http://www.atmos.umd.edu/~ocean/index_files/soda3.3.1_mn_download.htm, the GODAS dataset at http://cfs.ncep.noaa.gov/cfs/godas/monthly/, the ERA5 and ERA-Interim datasets at https://cds.climate.copernicus.eu/cdsapp#!/dataset/reanalysis-era5-single-levels-monthly-means and https://apps.ecmwf.int/datasets/data/interim-mdfa/levtype=sfc/, the TropFlux dataset at https://incois.gov.in/tropflux/tf_products.jsp, the OAFlux dataset at ftp://ftp.whoi.edu/pub/science/oaflux/data_v3, the CERES dataset at https://ceres.larc.nasa.gov/products-info.php?product=EBAF, the SRB dataset at the NASA Langley Research Center Atmospheric Sciences Data Center NASA/GEWEX SRB Project, and the ISCCP-FH dataset at https://isccp.giss.nasa.gov/projects/flux.html. The GPCP and ERSSTv5 datasets are provided by the NOAA/OAR/ESRL PSD, Boulder, Colorado, USA, from their Web site at https://www.esrl.noaa.gov/psd/. Source data are provided with this paper.

## Code availability

The codes used in this study to produce the data analyzed are available on a GitHub repository upon reasonable request to M.H.

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

## Acknowledgements

M.H. was supported by JSPS Overseas Research Fellowships 201860671. F.F.J. was supported by U.S. NSF grant AGS-1813611 and Department of Energy grant. DE-SC0005110. This is SOEST publication 11111 and IPRC contribution 1464.

## Author contributions

M.H. designed the study, collected and analyzed data, and wrote the manuscript; F.F.J. and M.F.S. discussed the results and implications and edited the paper.

## Competing interests

The authors declare no competing interests.
