## [Peer Review File · Nature Communications]

Reviewers' comments, first round -

Reviewer #1 (Remarks to the Author):

The authors have analyzed the ENSO asymmetry, the upper ocean heat budget—in particular—the nonlinear dynamic heating (NDH) as they call it, and the projected tropical Pacific climate changes in response to greenhouse gas forcing in the CMIP5 and CMIP6 modes. Largely through correlation analysis, they appear to have made the following major claims

- (1) subsurface NDH is a key controlling factor for ENSO asymmetry
- (2) weak NDH stems from the deficiencies in the dynamic ocean-atmosphere coupling
- (3) Future tropical (mean) climate response to greenhouse gas forcing is related to the response in the amplitude of ENSO

These claims are of interest to others in the field of climate dynamics. So the issues are how novel these claims are. Claim (1) does not appear to be a novel claim. Maybe I have missed something. Has claim (1) already been made in the following paper?

Hayashi, M & Jin, F.F. Subsurface nonlinear dynamical heating and ENSO asymmetry. *Geophys. Res. Lett.* 44, 12427-12345 (2017).

Whether Claim (3) is entirely novel appears to be questionable too. Is it already implied strongly in the study of the rectification effect of ENSO into the mean state of climate. Two papers are provided below so that the authors can quickly judge by themselves.

Liang, J., X.Q. Yang, and D.-Z. Sun The effect of ENSO events on the Tropical Pacific Mean Climate: Insights from an Analytical Model. *J. Climate*, 25, 7590-7606
Sun, D.-Z., T. Zhang, Y. Sun, and Y. Yu, 2014: Rectification of El Niño-Southern Oscillation into Climate Anomalies of Decadal and Longer Time-scales: Results from Forced Ocean GCM Experiments. *J. Climate*, 27, 2545-2561.

Claim (2) appears to be new, to the best knowledge of this reviewer. What can be questioned is the evidence presented for this claim, The evidence is given by Fig. 2 (c and d) which show a correlation between NDH and the two quantities related to the dynamical coupling between the atmosphere and ocean. But correlation is not causality. All of them can be a consequence of a third player. The same concerns apply to the evidence presented in this paper to support the paper's other two claims—claim (1) and claim (2). Note also that in the aspect of ENSO asymmetry, the spread among the models is small—most models have symmetric ENSO. With such a narrow spread of ENSO asymmetry among the majority of models and the existence of a few outliers, how much we really can infer from a linear correlation analysis about the existence or the lack of thereof a relationship?

The most novel and robust finding from this study, as this reviewer sees it, is the finding that the weak ENSO asymmetry, a common problem in CMIP5 models, remains a common problem in CMIP6 models. This underscores again the question why it is so difficult to simulate ENSO asymmetry. I agree with the authors in that ENSO asymmetry is a signal that has increasingly compelled us to look at the nonlinearity in the coupled tropical ocean atmosphere system. The authors are clearly doing that in this paper. Perhaps, what has prevented this review to see all the originality and importance of the results presented in this paper is that the scope of the paper is too broad. A too broad scope with many claims usually leave each claim weak or appearing to be repeating earlier claims with expanded data or in a different setting. The paper may also be improved by discussing the closely related papers more explicitly. In addition to the three papers that I have already mentioned, the following paper that apparently has some overlapping results:

Sun, Y., F. Wang, and D.-Z. Sun, 2016: Weak ENSO asymmetry due to weak nonlinear air-sea interaction in CMIP5 climate models. *Adv. Atmos. Sci.*, 33(3), 352-364.

The authors has put a great emphasis on the subsurface, but did not cite the following paper, probably the first paper to look at ENSO asymmetry in the subsurface:

Zhang, T., D.-Z. Sun, R. Neal, and P. Rasch, 2009: An Evaluation of ENSO Asymmetry in the Community Climate System Models: A View from the Subsurface. *J. Climate*, 22, 5933-5961.

Another highly relevant paper that authors have overlooked is

Liang, J., X.-Q. Yang, and D.-Z. Sun, 2017: Factors Determining the Asymmetry of ENSO. *J. Climate*, 30, 6097-6106.

This paper provides an example showing how weak NDH and weak ENSO asymmetry can be both a consequence of other erroneous factors, such as an excessive cold-tongue.

Individually dissected, the novelty or robustness of each claim has been questioned by this reviewer (who is probably among the more critical reviewers). The amount of work is clearly impressive. Getting the upper ocean heat budget done for all the CMIP5 and CMIP6 models is not a trivial task. Collectively, the results presented in this paper do add to our knowledge on issues why the models have a weak ENSO asymmetry and whether the changes in ENSO and the changes in the mean climate state are closely related. Perhaps narrowing the scope of the paper, reorganizing the results somewhat to highlight the most important result or the collective message, or making better connection with the missed literature to better illustrate the novelty of the present claims, may give this reviewer a stronger impression.

De-Zheng Sun

Reviewer #2 (Remarks to the Author):

This study provides a dynamical explanation for CMIP5 and CMIP6 models inability to simulate the observed asymmetry between El Nino and La Nina events. The authors identify a relationship between ENSO asymmetry and subsurface nonlinear dynamical heating. It is important to increase our understanding of how models project future tropical warming patterns - this study is a very relevant contribution to this effort and I recommend that it is suitable for publication after some comments are addressed.

General comments:

1. The authors identify a too-weak wind response to SST anomalies as a deficiency in the models. This was also identified as a deficiency in a recent study by Seager et al (2019), who also stated that CMIP5 models do not capture the observed shoaling of the thermocline in recent decades (and associated observed La Nina like warming pattern). Given that the authors find that the majority of H models project a deepening thermocline and El Nino-like warming pattern, there should be some discussion on this apparent inconsistency?

Seager, R., Cane, M., Henderson, N. et al. Strengthening tropical Pacific zonal sea surface temperature gradient consistent with rising greenhouse gases. *Nat. Clim. Chang.* 9, 517-522 (2019). <https://doi.org/10.1038/s41558-019-0505-x>

2. In the abstract and opening sentence, the cold tongue mean state is mentioned. As the long standing cold tongue bias was also identified as a shortcoming in Seager et al 2019 and has been found to be somewhat improved in CMIP6 (Grose et al 2020), it would be good if the authors can comment on whether the H-group models have a reduced cold tongue bias.

Grose, M et al. Insights from CMIP6 for Australia's future climate. *Earth's Future*

<https://www.essoar.org/doi/10.1002/essoar.10501525.1>

3. The NDH is evaluated in the eastern Pacific region but in Fig 2 is compared to wind stress anomalies in the central Pacific region - what is the reason for this, and what is the relationship between NDH and wind stress anomalies in the eastern Pacific region?

Minor comments:

L17: Opening sentence is somewhat unclear. "have" -> "has"?

L38: "despite that" -> "despite the fact that"

L43-45: "Thus, models fail badly in simulating observed ENSO skewness at about 1, with no improvement in CMIP6 compared to earlier phases" - this sentence is unclear.

L140: Do you mean Fig 4, not Supp Fig 7?

Fig 2: CP not defined

Fig 2 & 3: It is not clear what is denoted by the black circles and triangles.

Reviewer #3 (Remarks to the Author):

Review of "Dynamics for El Niño-La Niña asymmetry constrain equatorial Pacific warming pattern" by Hayashi et al.

Recommendation: Minor Revisions

Summary: This study investigates the relation between ENSO asymmetry and the equatorial Pacific warming pattern and finds out that half of the CMIP models strongly underestimate the subsurface nonlinear dynamical heating (NDH) due to underestimated wind-SST and heat flux-SST feedbacks. The models with high NDH show a strong relation between the ENSO amplitude change and the change in mean NDH. Further do these models show a strong relation between the ENSO amplitude change and the "El Niño-likeness" of the equatorial Pacific warming pattern, while the low NDH models show no clear relation.

Overall opinion: This is a very interesting, well structured and well elaborated study with convincing results, that gives interesting new results to the long discussed question of the ENSO amplitude change and equatorial Pacific warming pattern under global warming. The methodology is well described and it was easy to follow the text and the argumentation. I suggest to publish this study in Nature Communications after some minor revisions.

Minor comments:

26 & 84: You should also cite here Bayr et al. (2019)

49f: "However, the question of what dominant nonlinear dynamical process is causing it remains elusive." I in general agree with the authors that the dominant nonlinear dynamical process remains elusive, but there are several indications that non-linearities in the wind and shortwave feedback also play an important role in ENSO asymmetry (Frauen and Dommenges 2010; Lloyd et al. 2012; Bellenger et al. 2014; Karamperidou et al. 2017) and that the non-linearity of these feedbacks is more pronounced, if these feedbacks are stronger (Bayr et al. 2018). This should be discussed somewhere in the paper.

140: "Supplementary Fig. 7" => I think this should be "Supplementary Fig. 6"

176ff: "Therefore, enhancing the subsurface rectification effect that warms the eastern-Pacific subsurface temperature, deepens the mean-state thermocline, and thereby increases eastern-Pacific SST can lead to a more pronounced "El Niño-like" global-warming pattern, whereas suppressing the rectification effect leads to less eastern Pacific warming." This is a very interesting result, that opens the question, why some models simulate an increase in NDH and some models a decrease in NDH under global warming. Maybe this is beyond the scope of this paper, but my first guess would be that this could be related to wind feedback changes, with a similar mechanism as described in Bayr et al. (2020) by altering the atmospheric mean state.

Supplementary Table 1: Please indicate in this table in which sub-ensemble the models are, so that future studies could refer to this table and focus their research e.g. only on models with a high NDH.

References:

- Bayr T, Dommenges D, Latif M (2020) Walker Circulation controls ENSO Atmospheric Feedbacks in Uncoupled and Coupled Climate Model Simulations. *Clim Dyn*. <https://doi.org/10.1007/s00382-020-05152-2>
- Bayr T, Latif M, Dommenges D, et al (2018) Mean-state dependence of ENSO atmospheric feedbacks in climate models. *Clim Dyn* 50:3171–3194. <https://doi.org/10.1007/s00382-017-3799-2>
- Bayr T, Wengel C, Latif M, et al (2019) Error compensation of ENSO atmospheric feedbacks in climate models and its influence on simulated ENSO dynamics. *Clim Dyn* 53:155–172. <https://doi.org/10.1007/s00382-018-4575-7>
- Bellenger H, Guilyardi E, Leloup J, et al (2014) ENSO representation in climate models: From CMIP3 to CMIP5. *Clim Dyn* 42:1999–2018. <https://doi.org/10.1007/s00382-013-1783-z>
- Frauen C, Dommenges D (2010) El Niño and La Niña amplitude asymmetry caused by atmospheric feedbacks. *Geophys Res Lett* 37:L18801. <https://doi.org/10.1029/2010GL044444>
- Karamperidou C, Jin FF, Conroy JL (2017) The importance of ENSO nonlinearities in tropical pacific response to external forcing. *Clim Dyn* 49:2695–2704. <https://doi.org/10.1007/s00382-016-3475-y>
- Lloyd J, Guilyardi E, Weller H (2012) The role of atmosphere feedbacks during ENSO in the CMIP3 models. Part III: The shortwave flux feedback. *J Clim* 25:4275–4293. <https://doi.org/10.1175/JCLI-D-11-00178.1>

Response to reviewers' comments

Major changes from the original submission

1. We changed the overall format of the manuscript to fit the *Nature Communications* article as the original submission had been a transferred manuscript from *Nature Climate Change*.
2. We reorganized the Results section into three subsections and added a paragraph (Lines 56-72) to the end of the Introduction to properly represent the aims of this study. Especially, the first paragraphs of each subsection in Results highlight the purpose of our analysis.
3. We added a schematic figure (Fig. 3) that explains the physical mechanism for the relationship between the subsurface NDH and ENSO asymmetry (Introduction) and also the model biases therein (Results). The figure is sketched based on data analysis shown in Supplementary Figs. 2 and 4.
4. Possible roles of atmospheric nonlinearity on ENSO asymmetry bias are now discussed explicitly based on new data analysis (Lines 95-109). We replaced Figs. 2c and 2d of the original submission with new analysis for the atmospheric nonlinearities associated with ENSO's key feedback processes (Fig. 2c,d). Data in the original Fig. 2c,d is now used for Fig. 5, which shows the model biases associated with too-weak subsurface NDH.
5. Figures 3a-f and 3g,h of the original submission are now separated into two figures: Figure 4 shows the composites of ENSO asymmetry and mean NDH in each CMIP subgroup (L, H, HH) while scatterplots in Fig. 6 show the future changes of ENSO amplitude, mean NDH, and "El Niño-likeness" of the tropical warming pattern.
6. We merged Fig. 4 and Supplementary Fig. 7 of the original submission into Fig. 7, which includes the linear trends in both groups H and L so that the main purpose of this analysis could be clarified. Accordingly, the relevant two paragraphs about composited tropical warming in group H and L were merged (Lines 215-232). We also added the equatorial temperature trends in Fig. 7 (contours), which had been missed in the original submission by mistake. To focus on the temperature trends due to ENSO amplitude change, the precipitation trends in Fig. 4 in the original submission were moved to Supplementary Fig. 7.
7. We covered the previous literature on ENSO nonlinearity and rectification effect as much as possible as the *Nature Communications* article format allows up to 70 references.

Reply to reviewer #1

In this reply, blue font indicates the original comments from the reviewer. Line numbers and pages refer to the revised version of the manuscript. We assigned numbers **C1.1–4** for specific comments and **A1.1–4** for the corresponding answers for convenience. Please refer to “**Major changes from the original submission**” for overall changes in the present manuscript.

Overall reply:

Reviewer #1 (Remarks to the Author):

The authors have analyzed the ENSO asymmetry, the upper ocean heat budget—in particular—the nonlinear dynamic heating (NDH) as they call it, and the projected tropical Pacific climate changes in response to greenhouse gas forcing in the CMIP5 and CMIP6 modes. Largely through correlation analysis, they appear to have made the following major claims

- (1) subsurface NDH is a key controlling factor for ENSO asymmetry
- (2) weak NDH stems from the deficiencies in the dynamic ocean-atmosphere coupling
- (3) Future tropical (mean) climate response to greenhouse gas forcing is related to the response in the amplitude of ENSO

These claims are of interest to others in the field of climate dynamics. So the issues are how novel these claims are. Claim (1) does not appear to be a novel claim. Maybe I have missed something. Has claim (1) already been made in the following paper?

Hayashi, M & Jin. F.F. Subsurface nonlinear dynamical heating and ENSO asymmetry. *Geophys. Res. Lett.* 44, 12427-12345 (2017).

Whether Claim (3) is entirely novel appears to be questionable too. Is it already implied strongly in the study of the rectification effect of ENSO into the mean state of climate. Two papers are provided below so that the authors can quickly judge by themselves.

Liang, J., X.Q. Yang, and D.-Z. Sun 2012: The effect of ENSO events on the Tropical Pacific Mean Climate: Insights from an Analytical Model. *J. Climate*, 25 , 7590-7606

Sun, D.-Z., T. Zhang, Y. Sun, and Y. Yu, 2014: Rectification of El Nino-Southern Oscillation into Climate Anomalies of Decadal and Longer Time-scales: Results from Forced Ocean GCM Experiments. *J. Climate*, 27 , 2545-2561.

Claim (2) appears to be new, to the best knowledge of this reviewer. What can be questioned is the evidence presented for this claim, The evidence is given by Fig. 2 (c and d) which show a correlation between NDH and the two quantities related to the dynamical coupling between the atmosphere and ocean. But correlation is not causality. All of them can be a consequence of a third player. The same concerns apply to the evidence presented in this paper to support the paper's other two claims—claim (1) and claim (2).

Note also that in the aspect of ENSO asymmetry, the spread among the models is small—most models have symmetric ENSO. With such a narrow spread of ENSO asymmetry among the majority of models and the existence of a few outliers, how much we really can infer from a

linear correlation analysis about the existence or the lack of thereof a relationship?

We would like to thank Dr. Sun (reviewer #1) for providing these stimulating and very constructive comments. To best address the various questions and comments, we have summarized the reviewer's remarks into three main points: (I) the novelty of this study, (II) concern about the correlation analysis, and (III) lack of references. We first answer to each point generally and then provide detailed point-by-point replies to the other comments further below.

(I) Novelty of this study

We first would like to discuss if the suggested claims (1)–(3) above correctly represent this study. The main purpose of this study is to evaluate the model's ability to simulate ENSO nonlinear dynamics in CMIP5 and CMIP6 and the implications of correctly simulating ENSO for projecting future climate change in the tropics. Here we showed that (i) the majority of climate models still fail to reproduce ENSO asymmetry and that (ii) the subsurface NDH efficiency explains ~60% of the inter-model variance of simulated ENSO asymmetry.

Therefore, to be accurate, we would correct Claim (1) as follows: *“Subsurface NDH is a key controlling factor for the inter-model variance of ENSO asymmetry bias in the current generation of climate models”*. Our results provide valuable information for future improvement of climate models that have not been yet reported in previous studies nor in Hayashi and Jin (2017), who first reported the observed connection between the subsurface NDH and ENSO asymmetry in reanalysis datasets.

As for Claim (3), we would like to clarify that we are not arguing that we newly found the tropical mean climate response to ENSO amplitude change. As pointed out by the reviewer, this has been already suggested in many previous studies. These include, but are not limited to, Battisti and Hirst (1989), Jin et al. (2003), Sun and Zhang (2006), Liang et al. (2012), and Sun et al. (2014). Instead, we argue here that the tropical climate response to ENSO amplitude change can be detected only once the CMIP climate model ensemble is sternly constrained by the fidelity of simulated ENSO nonlinear dynamics. This is because ~70% of the CMIP models fail to produce ENSO nonlinear dynamics [see also (II) below]. Indeed, we found that the tropical warming pattern is not related to ENSO amplitude change at all in the climate models that do not simulate ENSO nonlinear dynamics well. Therefore, we would correct Claim (3) as follows: *“Only if climate models simulate ENSO nonlinear dynamics correctly, future tropical (mean) climate response to greenhouse gas forcing is related to the response in the amplitude of ENSO”*. We have clarified this point in the revised discussion (Lines 233-247), and also, we have added the previous literature about ENSO's nonlinear rectification effect to the main text (e.g., Sun and Zhang 2006; Liang et al. 2012; Sun et al. 2014).

The second claim (2), which was new to the reviewer, is indeed an important implication for improving the next generation of climate models. Ideally, one would want to reveal the

specific model parametrization (or “a third player”) that is the root cause of all the systematically biased processes detected in our data analysis. However, eventually, we could not find any reliable evidence for singling out a root cause for these model biases, leaving this important question open. Instead, we were able to highlight the important role that subsurface ocean dynamics play for simulating ENSO asymmetry. Please see (II) and A1.3 below.

In summary, this study does not aim to argue the claims (1)–(3) suggested by the reviewer. We apologize that the previous version of our manuscript was not sufficiently clear in pointing out the main conclusions. The novelty of this study exists in the following findings: (1) the majority of climate models in CMIP5 and CMIP6 cannot simulate ENSO’s asymmetry and nonlinear dynamics correctly; (2) the subsurface NDH is a key controlling factor for the inter-model variance of ENSO asymmetry bias in climate models; (3) the climate impact of ENSO change can be detected only once the climate model ensemble is constrained by the fidelity to simulated ENSO nonlinear dynamics. We reorganized the Results section and clarified the discussion of these points throughout the revised manuscript.

(II) Concern about the correlation analysis

Next, we would like to address the concern that the correlation analysis shown in Fig. 2(c,d) of the original submission may not guarantee causality. The reviewer points out that all of them (i.e., NDH and the two quantities related to the dynamical coupling between the atmosphere and ocean) can be a consequence of a third player. The original Fig. 2c intends to show that the linear dynamic coupling from the atmosphere to the ocean associated with ENSO’s SST is too weak in all the models that have weak subsurface NDH. Also, our main message from the original Fig. 2d is the serious bias of the models in generating a response of the EUC, which plays the major role in generating subsurface NDH, to the zonal wind stress. Thus, the levels of correlation coefficients in these panels themselves are not critical for our conclusions. In the revised manuscript, these two panels have been removed but replaced with boxplots in new Fig. 5, highlighting the model biases in these processes (Lines 129-155).

Although there may exist a possible factor that may control all of the biases, it is reasonable to think that a realistic EUC variability comparable with observations cannot be generated mainly because of two reasons: (1) the atmospheric dynamic forcing anomaly is too weak and (2) also the forcing momentum cannot produce the subsurface ocean current realistically. This leaves us with two open questions: (1) why is the wind stress response to ENSO SSTA too weak and (2) what prevents a realistic EUC response to a given wind forcing. The latter is a new question in this context to our knowledge. We made these questions open explicitly (Lines 164-167).

Another question from the reviewer about the correlation analysis is: “With such a narrow spread of ENSO asymmetry among the majority of models and the existence of a few outliers, how much we really can infer from a linear correlation analysis about the existence or the lack of thereof a relationship?” It is true that only ~30% of the CMIP models can produce both the ENSO asymmetry and realistic subsurface NDH to a certain extent. We

think that this result itself is important and a critical caution that needs to be taken into account in future studies on the ENSO response to global warming. As reviewer #1 commented [C1.1], this is a major message from this study. As for the correlation analysis, it is of course ideal to have more independent samples that have higher ENSO asymmetry. However, the data in Fig. 2b, at least, shows the fact that there are only a few models that have subsurface NDH efficiency and ENSO asymmetry comparable to reanalysis (Line 88). There could be a way to increase the number of samples, such as perturbed physics experiments using some group H models, which may allow us to examine contribution of each nonlinear process to ENSO asymmetry, but regrettably this is beyond the scope of this study. Nevertheless, considering the fact that the simulated atmospheric nonlinearity is already comparable with observational datasets (see A1.1 and new Fig. 2c,d), it is worth suggesting a new possibility to explain the lack of ENSO asymmetry in climate models.

(III) Lack of references

We apologize that the original submission did not fully cover the previous literature about the ENSO nonlinearity (e.g., Zhang et al. 2009; Y. Sun et al. 2016; Liang et al. 2017) and ENSO's mean state rectification effects (e.g., Liang et al. 2012; D.-Z. Sun et al. 2014). Since the current "Article" format of *Nature Communications* has more space than the previous "Letter" format of Nature journals, we extended the references and introductory section in the revised version. We revised both the Introduction and Results sections and added a paragraph with extra data analysis for other possible sources for ENSO nonlinearity (Lines 95-109). We appreciate the reviewer for providing several important articles that were missed in the original submission. We also added Sun and Zhang (2006), Sun et al. (2006), and other relevant articles. We believe this change sharpens the scope of this study and highlights our findings in the context of the previous literature [see (I) above]. Please see also A1.1–3 below.

Replies to the specific comments:

[C1.1] The most novel and robust finding from this study, as this reviewer sees it, is the finding that the weak ENSO asymmetry, a common problem in CMIP5 models, remains a common problem in CMIP6 models. This underscores again the question why it is so difficult to simulate ENSO asymmetry. I agree with the authors in that ENSO asymmetry is a signal that has increasingly compelled us to look at the nonlinearity in the coupled tropical ocean atmosphere system. The authors are clearly doing that in this paper. Perhaps, what has prevented this review to see all the originality and importance of the results presented in this paper is that the scope of the paper is too broad. A too broad scope with many claims usually leave each claim weak or appearing to be repeating earlier claims with expanded data or in a different setting. The paper may also be improved by discussing the closely related papers more explicitly. In addition to the three papers that I have already mentioned, the following paper that apparently has some overlapping results:

Sun, Y., F. Wang, and D.-Z. Sun, 2016: Weak ENSO asymmetry due to weak nonlinear air-sea interaction in CMIP5 climate models. *Adv. Atmos. Sci.*, 33(3), 352-364.

[A1.1] We thank the reviewer for underscoring the finding of this study. As mentioned in the

overall reply (I) above, we have changed the Introduction and discussion to sharpen the scope of this study. The missing previous literature about ENSO asymmetry bias and ENSO's nonlinear rectification effect has been added to the Introduction and Results sections (e.g., Lines 50-55, 56-72, 95-109).

As for the atmospheric nonlinearity, we have added an extra analysis to a new paragraph at Lines 95-109 for explicitly discussing the closely related papers such as Y. Sun et al. (2016). Figure R1.1 (Fig. 2c,d) shows ENSO SST asymmetry as functions of the atmospheric nonlinearities measured by the skewness of anomalies of the central-Pacific zonal wind stress and the Niño-3+4 net surface heat flux. Regardless of the fact that the majority of the CMIP models have very poor Niño-3 SST anomaly skewness (Fig. 2a), the atmospheric nonlinearity is captured or overrepresented by the CMIP models overall. The right panel of Figure R1.1 (Fig. 2c) shows the correlation between the wind-stress skewness and ENSO asymmetry. Although ENSO asymmetry tends to increase with respect to the wind-stress skewness, the observational datasets (red) appear outside of the correlated relationship for the CMIP models (green and blue). In other words, many models have a wind skewness comparable with reanalysis, but ENSO asymmetry is poorly simulated. **This indicates that increasing the atmospheric nonlinearity alone cannot improve the ENSO SST asymmetry, calling a need of other sources for the nonlinearity.** Note that we do not say that the ocean nonlinear process is the most important to explain the ENSO asymmetry and we do not rule out the importance of atmospheric nonlinearity to SST anomaly.

Figure R1.1 | ENSO SST skewness as functions of skewness of anomalies of the zonal wind stress in the central-Pacific (CP) domain and the net surface heat flux in the Niño-3 and Niño-4 regions (left). Please see Fig. 2c,d in the main text.

[C1.2] The authors has put a great emphasis on the subsurface, but did not cite the following paper, probably the first paper to look at ENSO asymmetry in the subsurface:

Zhang, T., D.-Z. Sun, R. Neal, and P. Rasch, 2009: An Evaluation of ENSO Asymmetry in the Community Climate System Models: A View from the Subsurface. *J. Climate*, 22, 5933-5961.

[A1.2] Thank you for introducing a paper that shows the asymmetric feature of ENSO in the equatorial upper ocean between its warm and cold phases. We cite the article (ref.34) in the revised manuscript at Lines 58, for instance, as a supporting evidence of subsurface temperature asymmetry of ENSO. We also added Sun & Zhang (2006) and Sun et al. (2014) that show the subsurface nonlinearity of ENSO.

[C1.3] Another highly relevant paper that authors have overlooked is Liang, J., X.-Q. Yang, and D.-Z. Sun, 2017: Factors Determining the Asymmetry of ENSO. J. Climate, 30, 6097-6106.

This paper provides an example showing how weak NDH and weak ENSO asymmetry can be both a consequence of other erroneous factors, such as an excessive cold-tongue.

[A1.3] We thank the review for providing this reference. The model used in the above paper is a low-order model that includes nonlinear advective effects in the ocean surface mixed layer. However, the model does not implement ocean dynamics below the ocean mixed layer, which is the focus of the present work. We showed a significant connection between ENSO SST asymmetry and subsurface NDH in climate models but also showed that the mixed-layer NDH in our study was not able to explain the simulated weak ENSO SST asymmetry in the climate models (Lines 89-90). The latter was also implied in the paper above, thus we added a discussion of the study in Lines 90-92.

Motivated by a comment by reviewer #2 [C2.2], we analyzed a mean-state cold-tongue bias index (CTI; Grose et al. 2020) for assessing the possible relationship between ENSO asymmetry and an excessive cold tongue mean state. As seen in Figure R1.2 below, we find no correlation among CMIP climate models between the CTI and subsurface NDH efficiency nor ENSO asymmetry. Thus, we do not yet have evidence to indicate a role of excessive cold tongue, nor other systematic model biases, for the simulated ENSO asymmetry and subsurface NDH biases. See also the overall replies (II).

Figure R1.2 | Cold-tongue bias index and its relationships with the subsurface NDH efficiency and ENSO asymmetry (i.e., Niño-3 SST anomaly skewness). Shown are the historical simulations

of CMIP5 and CMIP6 models (blue) and the multi-reanalysis products (red) (see Supplementary Table 1 for data availability). The subsurface NDH efficiency (left) and ENSO asymmetry (right) are the same as in Fig. 2b. The x-axis represents cold tongue bias measured by the cold-tongue index (CTI; Grose et al. 2020), defined by the mean SST over 155-175°E and 10°S-10°N relative to the multi-reanalysis mean (see Supplementary Fig. 3).

[C1.4] Individually dissected, the novelty or robustness of each claim has been questioned by this reviewer (who is probably among the more critical reviewers). The amount of work is clearly impressive. Getting the upper ocean heat budget done for all the CMIP5 and CMIP6 models is not a trivial task. Collectively, the results presented in this paper do add to our knowledge on issues why the models have a weak ENSO asymmetry and whether the changes in ENSO and the changes in the mean climate state are closely related. Perhaps narrowing the scope of the paper, reorganizing the results somewhat to highlight the most important result or the collective message, or making better connection with the missed literature to better illustrate the novelty of the present claims, may give this reviewer a stronger impression.

De-Zheng Sun

[A1.4] We again thank Dr. Sun for providing helpful comments that significantly assisted in improving our manuscript and clarifying our arguments.

Reply to reviewer #2

In this reply, blue font indicates the original comments from the reviewer. Line numbers and pages refer to the revised version of the manuscript. We assigned numbers C2.1–9 for comments and A2.1–9 for the corresponding answers for convenience. Please refer to “**Major changes from the original submission**” for overall changes in the present manuscript.

Overall reply:

Reviewer #2 (Remarks to the Author):

This study provides a dynamical explanation for CMIP5 and CMIP6 models inability to simulate the observed asymmetry between El Nino and La Nina events. The authors identify a relationship between ENSO asymmetry and subsurface nonlinear dynamical heating. It is important to increase our understanding of how models project future tropical warming patterns - this study is a very relevant contribution to this effort and I recommend that it is suitable for publication after some comments are addressed.

We thank the reviewer for providing very helpful comments.

Replies to the specific comments:

General comments:

[C2.1] 1. The authors identify a too-weak wind response to SST anomalies as a deficiency in the models. This was also identified as a deficiency in a recent study by Seager et al (2019), who also stated that CMIP5 models do not capture the observed shoaling of the thermocline in recent decades (and associated observed La Nina like warming pattern). Given that the authors find that the majority of H models project a deepening thermocline and El Nino-like warming pattern, there should be some discussion on this apparent inconsistency?

Seager, R., Cane, M., Henderson, N. et al. Strengthening tropical Pacific zonal sea surface temperature gradient consistent with rising greenhouse gases. *Nat. Clim. Chang.* 9, 517–522 (2019). <https://doi.org/10.1038/s41558-019-0505-x>

[A2.1] We thank the reviewer for this helpful comment. We would not say that the majority of H models project a deepening thermocline and El Nino-like warming pattern; rather, we report that the projected tropical warming pattern can be constrained by ENSO amplitude change in group H models. In contrast, we also found that *the group L models, having a too-weak wind response to SST anomalies, project an “El Niño-like” tropical warming pattern regardless of ENSO amplitude change*. This is consistent with the reviewer’s comment that *CMIP5 models that have a too-weak wind response to SST anomalies do not capture the La Nina like warming pattern*. We have added a comment about this consistency in the Results section about the tropical response in group L (Lines 240-242). We appreciate the reviewer for providing this supportive information.

[C2.2] 2. In the abstract and opening sentence, the cold tongue mean state is mentioned. As the long standing cold tongue bias was also identified as a shortcoming in Seager et al 2019 and has been found to be somewhat improved in CMIP6 (Grose et al 2020), it would be good if the

authors can comment on whether the H-group models have a reduced cold tongue bias.
 Grose, M et al. Insights from CMIP6 for Australia's future climate. Earth's Future
<https://www.essoar.org/doi/10.1002/essoar.10501525.1>

[A2.2] We appreciate reviewer #2 for introducing a new interesting paper that evaluated a mean-state cold-tongue bias index (CTI; Grose et al. 2020) in CMIP5 and CMIP6. We followed the paper to assess the relationship between ENSO asymmetry and CTI using the climatology of the historical simulations relative to the multi-reanalysis average (see Figure R2.1 below). There is an improvement in terms of CTI from the group L to H models (this is interesting and mentioned in the main text with Supplementary Fig. 3 at Lines 132 and 148-149); however, the difference between the two is not statistically significant at 95% and the group HH models do not show any further improvement from group H. Thus, we cannot yet conclude that the cold tongue bias was improved in the models that have higher NDH efficiency and ENSO asymmetry. Please see also A1.3 (Figure R1.2) in the reply to reviewer #1.

Figure R2.1 | Inter-quartile ranges and maximum/minimum levels of cold-tongue index (CTI: climatological SST bias averaged over 155-175°E and 10°S-10°N; see Grose et al. 2020) in CMIP historical simulations (groups L, H, and HH). The bias is defined as the difference from the multi-reanalysis mean used in this study. See Supplementary Fig. 3.

[C2.3] 3. The NDH is evaluated in the eastern Pacific region but in Fig 2 is compared to wind stress anomalies in the central Pacific region - what is the reason for this, and what is the relationship between NDH and wind stress anomalies in the eastern Pacific region?

[A2.3] We thank the reviewer for the question. The two boxes are selected for capturing the centers of action for ENSO-associated NDH and wind stress variability, respectively. As shown in Supplementary Figure 4, the westerly wind stress anomaly associated with ENSO, which is the key element for determining ENSO instability, is most pronounced over **the central Pacific region (150°E-120°W)**. The westerly wind anomaly corresponds to the weakening of

climatological easterly trade wind that produces the climatological subsurface zonal current (EUC). The weakening of the EUC in the subsurface mainly determines the subsurface NDH, which in turn result in mean positive NDH and large NDH variability on **the eastern side of the dateline (180°-100°W)** (Fig. 1; see also Hayashi and Jin 2017). Therefore, we compared the “eastern Pacific NDH” and “central Pacific wind stress”. We explain this point in the new revised Introduction in a paragraph about the physical mechanisms associated with EUC, NDH and ENSO asymmetry (Lines 60-67).

The results (i.e., the increase of NDH with respect to wind stress, too-weak wind stress variability for ENSO) are not affected if we use the wind stress anomaly averaged over the Pacific domain that includes the eastern Pacific region (see Figure R2.2), which is also sometimes used for evaluating the wind response to ENSO SST anomalies. Note that Fig. 2c has been removed [please see the overall reply (II) to reviewer #1].

Figure R2.2 | Same plots as in Figure 2c of the original submission, but for the zonal wind stress anomalies averaged over the central Pacific domain (150°E-120°W; left) and the entire Pacific domain (120°E-80°W; right). Shown are the reanalysis datasets (red) and the CMIP5 and CMIP6 models (blue).

Minor comments:

[C2.4] L17: Opening sentence is somewhat unclear. "have" -> "has"?

[A2.4] We corrected it. Thank you.

[C2.5] L38: "despite that" -> "despite the fact that"

[A2.5] Corrected. Thank you.

[C2.6] L43-45: "Thus, models fail badly in simulating observed ENSO skewness at about 1, with no improvement in CMIP6 compared to earlier phases" - this sentence is unclear.

[A2.5] Thank you for the comment. We have changed the sentence to make its meaning clearer as follows: “Thus, CMIP climate models fail badly in simulating ENSO skewness that is about 1 in

observations, indicating no improvement in CMIP6 compared to earlier CMIP phases” (Lines 46-48)

[C2.7] L140: Do you mean Fig 4, not Supp Fig 7?

[A2.7] Thank you for pointing out this typo. It is corrected with Supplementary Fig. 7 (Supp. Fig. 6 of the original submission), which shows that **the weakening of ENSO is attributable to the declining oceanic wave response to equatorial-Pacific zonal winds due to an intensified thermal stratification** (Line 193-195) in the group H-wEN models.

[C2.8] Fig 2: CP not defined

[A2.8] Thank you. We defined it as “central-Pacific (CP)” in the revised caption.

[C2.9] Fig 2 & 3: It is not clear what is denoted by the black circles and triangles.

[A2.9] The captions for these figures were insufficient. The circles and triangles indicate the CMIP5 and CMIP6 models, respectively. We corrected the captions of Figs. 2 and 3. Thank you for the comment.

Reply to reviewer #3

In this reply, blue font indicates the original comments from the reviewer. Line numbers and pages refer to the revised version of the manuscript. We assigned numbers C3.1–5 for comments and A3.1–5 for the corresponding answers for convenience. Please refer to “**Major changes from the original submission**” for overall changes in the present manuscript.

Overall reply:

Reviewer #3 (Remarks to the Author):

Review of “Dynamics for El Niño-La Niña asymmetry constrain equatorial Pacific warming pattern” by Hayashi et al.

Recommendation: Minor Revisions

Summary: This study investigates the relation between ENSO asymmetry and the equatorial Pacific warming pattern and finds out that half of the CMIP models strongly underestimates the subsurface nonlinear dynamical heating (NDH) due to underestimated wind-SST and heat flux-SST feedbacks. The models with high NDH show a strong relation between the ENSO amplitude change and the change in mean NDH. Further do these models show a strong relation between the ENSO amplitude change and the “El Nino-likeness” of the equatorial Pacific warming pattern, while the low NDH models show no clear relation.

Overall opinion: This is a very interesting, well structured and well elaborated study with convincing results, that gives interesting new results to the long discussed question of the ENSO amplitude change and equatorial Pacific warming pattern under global warming. The methodology is well described and it was easy to follow the text and the argumentation. I suggest to publish this study in Nature Communications after some minor revisions.

We thank the reviewer for providing very helpful comments.

Replies to the specific comments:

Minor comments:

[C3.1] 26 & 84: You should also cite here Bayr et al. (2019)

[A3.1] We cite Bayr et al. (2019) in the revised version (ref.15). We thank the reviewer for suggesting the article about the atmospheric feedback errors.

[C3.2] 49f: “However, the question of what dominant nonlinear dynamical process is causing it remains elusive.” I in general agree with the authors that the dominant nonlinear dynamical process remains elusive, but there are several indications that non-linearities in the wind and shortwave feedback also play an important role in ENSO asymmetry (Frauen and Dommenget 2010; Lloyd et al. 2012; Bellenger et al. 2014; Karamperidou et al. 2017) and that the non-linearity of these feedbacks is more pronounced, if these feedbacks are stronger (Bayr et al. 2018). This should be discussed somewhere in the paper.

[A3.2] Thank you for the comment. As the review for the previous literature about ENSO nonlinearity was insufficient in the original submission, we have added new Fig. 2c,d and a subsection in the Results section to show the nonlinearity in the atmospheric variability (Lines 95-109). We also added the suggested references.

[C3.3] 140: “Supplementary Fig. 7” => I think this should be “Supplementary Fig. 6”

[A3.3] Thank you. It is corrected. In the revised version, it is changed to Supplementary Fig. 7 as new Supplementary Fig. 3 has been added.

[C3.4] 176ff: “Therefore, enhancing the subsurface rectification effect that warms the eastern-Pacific subsurface temperature, deepens the mean-state thermocline, and thereby increases eastern-Pacific SST can lead to a more pronounced “El Niño-like” global-warming pattern, whereas suppressing the rectification effect leads to less eastern Pacific warming.” This is a very interesting result, that opens the question, why some models simulate an increase in NDH and some models a decrease in NDH under global warming. Maybe this is beyond the scope of this paper, but my first guess would be that this could be related to wind feedback changes, with a similar mechanism as described in Bayr et al. (2020) by altering the atmospheric mean state.

[A3.4] As we have mentioned in the main text, it is still an open question whether ENSO amplitude, and thus mean NDH, increases or decreases in a future climate. Actually, we tried to solve this question. However, we only could identify that decreasing the ENSO amplitude and mean NDH **was attributable to the declining oceanic wave response to equatorial-Pacific zonal winds due to an intensified thermal stratification** (Lines 194-195; Supplementary Figure 7). We also examined the mean-state cold-tongue bias in the historical simulations [please see C2.2 and Figure R2.1 above], but it does not separate the ENSO amplitude change even in group H (e.g., both models “M” in H-wEN and “U” in H-sEN have too cold bias similarly).

We also plotted the zonal wind-stress feedback in the historical and future scenario simulations in group H (Figure R3.1 below). While the majority of group H-sEN (red) models show the increase of the wind feedback, the half models in group H-wEN (blue) also do. Thus, at this moment, we cannot conclude that the **wind feedback changes** contribute to the ENSO amplitude and NDH changes in our study.

Figure R3.1 | Central-Pacific zonal wind-stress feedback to Niño-3 SST anomaly ($0.01 \text{ N/m}^2/\text{K}$) in CMIP5 and CMIP6. The x-axis is for the historical simulation at each data period while the y-axis is for the future climate of RCP8.5 and SSP5-8.5 scenario simulations (2051-2100). Black line indicates $y=x$. Red and blue circles represent the strengthening (sEN) and weakening (wEN) of ENSO, respectively. Note that the detrended anomalies are used for calculating the regression coefficients (i.e., feedback).

[C3.5] Supplementary Table 1: Please indicate in this table in which sub-ensemble the models are, so that future studies could refer to this table and focus their research e.g. only on models with a high NDH.

[A3.5] We added a column “Group” for indicating in which sub-ensemble (L, H, HH) the models are. Thank you for the suggestion.

References:

- Bayr T, Dommenges D, Latif M (2020) Walker Circulation controls ENSO Atmospheric Feedbacks in Uncoupled and Coupled Climate Model Simulations. *Clim Dyn*. <https://doi.org/10.1007/s00382-020-05152-2>
- Bayr T, Latif M, Dommenges D, et al (2018) Mean-state dependence of ENSO atmospheric feedbacks in climate models. *Clim Dyn* 50:3171–3194. <https://doi.org/10.1007/s00382-017-3799-2>
- Bayr T, Wengel C, Latif M, et al (2019) Error compensation of ENSO atmospheric feedbacks in climate models and its influence on simulated ENSO dynamics. *Clim Dyn* 53:155–172. <https://doi.org/10.1007/s00382-018-4575-7>
- Bellenger H, Guilyardi E, Leloup J, et al (2014) ENSO representation in climate models: From CMIP3 to CMIP5. *Clim Dyn* 42:1999–2018. <https://doi.org/10.1007/s00382-013-1783-z>
- Frauen C, Dommenges D (2010) El Niño and La Niña amplitude asymmetry caused by atmospheric feedbacks. *Geophys Res Lett* 37:L18801. <https://doi.org/10.1029/2010GL044444>
- Karamperidou C, Jin FF, Conroy JL (2017) The importance of ENSO nonlinearities in tropical pacific response to external forcing. *Clim Dyn* 49:2695–2704.

<https://doi.org/10.1007/s00382-016-3475-y>

Lloyd J, Guilyardi E, Weller H (2012) The role of atmosphere feedbacks during ENSO in the CMIP3 models. Part III: The shortwave flux feedback. *J Clim* 25:4275–4293.

<https://doi.org/10.1175/JCLI-D-11-00178.1>

Reviewers' comments, second round -

Reviewer #1 (Remarks to the Author):

The authors have satisfactorily addressed the points raised in the previous round of review. The revised manuscript is suitable for publication.

De-Zheng Sun

Professor

Department of Atmospheric and Oceanic Sciences

Fudan University

No.2005 Songhu Road, Yangpu District, Shanghai 200438, China

Tel: +86-021-31248808

Reviewer #2 (Remarks to the Author):

The authors have done a large amount of work to respond to the reviewers comments. In my opinion the manuscript is suitable for publication.

Reviewer #3 (Remarks to the Author):

2. Review of "Dynamics for El Niño-La Niña asymmetry constrain equatorial Pacific warming pattern" by Hayashi et al.

Recommendation: Minor Revisions

The authors addressed all my comments to my satisfaction, except one.

Minor comments:

Fig. 2d): It is not a good way to represent the non-linearity of the heat flux feedback by the skewness of the heat flux over combined Nino3 + Nino4 region. Therefore I think the observations show such a weak non-linearity in Fig. 2d). From my experience the non-linearity of the heat flux feedback is largest in observations and underestimated in nearly all models. This should become more visible if you use e.g., the difference West - East box (140°E - 170°E, 5°S - 5°N) - (140°W - 170°W, 5°S - 5°N) of the difference between the composites of El Niño - La Niña, like shown in Fig. 14c in Bayr et al. (2018) for the short-wave feedback, which dominates the non-linearity of the heat flux feedback.

References:

Bayr T, Latif M, Dommenges D, et al (2018) Mean-state dependence of ENSO atmospheric feedbacks in climate models. *Clim Dyn* 50:3171-3194. <https://doi.org/10.1007/s00382-017-3799-2>

Responses to reviewers' comments

Please find our replies (black) to reviewers' comments (blue) below. In addition, we implemented several format and style changes throughout the text following the editor's guidance.

Reviewer #1 (Remarks to the Author):

The authors have satisfactorily addressed the points raised in the previous round of review. The revised manuscript is suitable for publication.

De-Zheng Sun
Professor
Department of Atmospheric and Oceanic Sciences
Fudan University
No.2005 Songhu Road, Yangpu District, Shanghai 200438, China
Tel: +86-021-31248808

Response: Thank you for your helpful feedback and comments during the review process.

Reviewer #2 (Remarks to the Author):

The authors have done a large amount of work to respond to the reviewers comments. In my opinion the manuscript is suitable for publication.

Response: Thank you for your helpful feedback and comments during the review process.

Reviewer #3 (Remarks to the Author):

2. Review of "Dynamics for El Niño-La Niña asymmetry constrain equatorial Pacific warming pattern" by Hayashi et al.

Recommendation: Minor Revisions

The authors addressed all my comments to my satisfaction, except one.

Response: Thank you for your helpful feedback and comments during the review process. Please find our reply to your comment below.

Minor comments:

Fig. 2d): It is not a good way to represent the non-linearity of the heat flux feedback by the skewness of the heat flux over combined Niño3 + Niño4 region. Therefore I think the observations show such a weak non-linearity in Fig. 2d). From my experience the non-linearity of the heat flux feedback is largest in observations and underestimated in nearly all models. This should become more visible if you use e.g., the difference West – East box (140°E – 170°E, 5°S – 5°N) – (140°W – 170°W, 5°S – 5°N) of the difference between the composites of El Niño – La Niña, like shown in Fig. 14c in Bayr et al. (2018) for the short-wave feedback, which dominates the non-linearity of the heat flux feedback.

References:

Bayr T, Latif M, Dommenges D, et al (2018) Mean-state dependence of ENSO atmospheric feedbacks in climate models. *Clim Dyn* 50:3171–3194. <https://doi.org/10.1007/s00382-017-3799-2>

Response: Thank you for this very helpful comment. We followed your suggestion and replaced the x-axis of the previous Fig. 2d with a zonal shortwave feedback contrast analysis (revised Fig. 2c). In agreement with Bayr et al. (2018), we can confirm a statistically significant moderate positive linear relationship ($r=0.37$, $p=0.008$) between the zonal contrast of equatorial Pacific shortwave (SW) anomalies and ENSO skewness in the models. However, the simulated ENSO skewness is generally too low even in the climate models that have higher feedback asymmetry (see revised Fig. 2c). We added these results and discussion thereof in the revised manuscript (lines 98-116).

We also changed the previous Fig. 2c, which showed the wind stress skewness and ENSO asymmetry. In the previous manuscript, Figure 2 (x-axis) presented the skewness values of the Niño-3+4 net surface heat flux- (**d**) and central-Pacific zonal wind stress (**c**) anomalies as one of the simplest ways to indicate the nonlinearity in two key atmospheric variabilities associated with ENSO's feedback processes. For consistency with the revised Fig. 2c (see the above response), the wind stress skewness has been replaced with the zonal wind feedback asymmetry (revised Fig. 2d). Indeed, enhancing the wind-SST coupling nonlinearity would not be an efficient way to improve the ENSO SST asymmetry, consistent with the previous Fig. 2c.

Figure 2c and 2d of the revised manuscript. ENSO SST skewness as functions of feedback asymmetry between positive and negative ENSO phases, which is the difference of the regression coefficients of the shortwave zonal contrast (c) and central-Pacific wind stress (d) anomalies onto the positive and negative Niño-3 SST anomalies. See the main text for details.